OBSERVATION

# Generation of Angiotensin-Converting Enzyme 2/Transmembrane Protease Serine 2-Double-Positive Human Induced Pluripotent Stem Cell-Derived Spheroids for Anti-Severe Acute Respiratory Syndrome Coronavirus 2 Drug Evaluation

Nobuyo Higashi-Kuwata,[a] Shigeharu G. Yabe,[b] Satsuki Fukuda,[b] Junko Nishida,[b] Miwa Tamura-Nakano,[c] Shin-ichiro Hattori,[a] Hitoshi Okochi,[b] Hiroaki Mitsuya[a,d]

[a]Department of Refractory Viral Diseases, National Center for Global Health and Medicine Research Institute, Tokyo, Japan
[b]Department of Regenerative Medicine, Research Institute, National Center for Global Health and Medicine, Tokyo, Japan
[c]Communal Laboratory, Research Institute, National Center for Global Health and Medicine, Tokyo, Japan
[d]Experimental Retrovirology Section, National Cancer Institute, NIH, Bethesda, Maryland, USA

**ABSTRACT** We newly generated two human induced pluripotent stem cell (hiPSC)-derived spheroid lines, termed Spheroids$_{-4M}$$^{ACE2-TMPRSS2}$ and Spheroids$_{-15M63}$$^{ACE2-TMPRSS2}$, both of which express angiotensin-converting enzyme 2 (ACE2) and transmembrane protease serine 2 (TMPRSS2), which are critical for severe acute respiratory syndrome coronavirus 2 (SARS-CoV-2) infection. Both spheroids were highly susceptible to SARS-CoV-2 infection, and two representative anti-SARS-CoV-2 agents, remdesivir and 5h (an inhibitor of SARS-CoV-2's main protease), inhibited the infectivity and replication of SARS-CoV-2 in a dose-dependent manner, suggesting that these human-derived induced spheroids should serve as valuable target cells for the evaluation of anti-SARS-CoV-2 activity.

**IMPORTANCE** The hiPSC-derived spheroids we generated are more expensive to obtain than the human cell lines currently available for anti-SARS-CoV-2 drug evaluation, such as Calu-3 cells; however, the spheroids have better infection susceptibility than the existing human cell lines. Although we are cognizant that there are human lung (and colonic) organoid models for the study of SARS-CoV-2, the production of those organoids is greatly more costly and time consuming than the generation of human iPSC-derived spheroid cells. Thus, the addition of human iPSC-derived spheroids for anti-SARS-CoV-2 drug evaluation studies could provide the opportunity for more comprehensive interpretation of the antiviral activity of compounds against SARS-CoV-2.

**KEYWORDS** COVID-19, SARS-CoV-2, TMPRSS2, hACE2, hiPSC-derived spheroids, *in vitro* drug evaluation

In the search for antiviral agents against certain human-pathogenic viruses, the use of human primary cells where the virus infects well and replicates well is desirable. However, a number of primary or long-term-cultured human-derived cells do not support their infection or replication well. Moreover, certain compounds, such as some nucleoside analogues (NAs), must be intracellularly triphosphorylated for them to exert their antiviral activity, while the efficiency of the anabolic phosphorylation differs depending on cell types (1, 2). Thus, it would be preferable if we could generate suitable human-derived host cells for each aimed virus, where the pathogenic virus in question exerts its infectivity and replicates well, using human induced pluripotent stem cells (hiPSC). However, drug evaluation using hiPSC requires the production of considerable numbers of hiPSC, and the induction process is rather complicated and costly (3, 4).

Address correspondence to Hiroaki Mitsuya, hiroaki.mitsuya2@nih.gov or hmitsuya@hosp.ncgm.go.jp, or Nobuyo Higashi-Kuwata, nkuwata@ri.ncgm.go.jp.

The authors declare no conflict of interest.

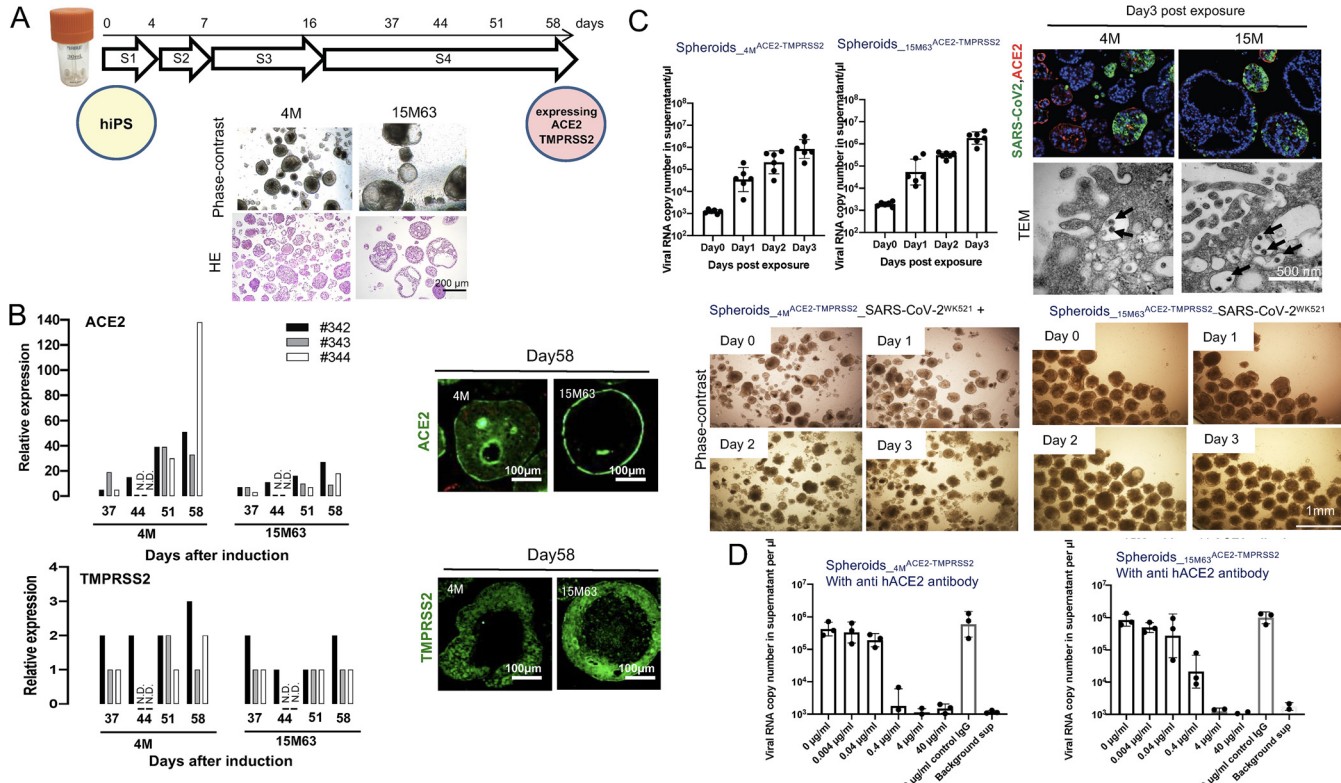

**FIG 1** Induction of Spheroids_4M^ACE-TMPRSS2 and Spheroids_15M63^ACE-TMPRSS2 from human induced pluripotent stem cells. To obtain large amounts of three-dimensional (3-D) spheroids for anti-SARS-CoV-2 assays, two hiPSC lines, TkDN4-M and 15M63, were differentiated in suspension culture using bioreactors. These hiPS cell lines were established from different donors. After confirming the expression levels of ACE2 and TMPRSS2, key factors for SARS-CoV-2 infection, the induced Spheroids^ACE-TMPRSS2 (Spheroids_4M^ACE-TMPRSS2 and Spheroids_15M63^ACE-TMPRSS2) were examined on day 58 postinduction for susceptibility to SARS-CoV-2 infection and the effect of anti-human ACE2 monoclonal antibody on their susceptibility to SARS-CoV-2. (A) Overview of the 4-stage differentiation protocol of hiPSC lines in suspension culture, and representative phase-contrast and hematoxylin-and-eosin (HE)-staining images of hiPSC-derived spheroids on day 58. S, stage. (B) Left, the mRNA expression levels of ACE2 and TMPRSS2 relative to the levels in the control (human lung tissue) were examined using RT-qPCR; right, immunocytostaining (in green) was performed for detecting ACE2 and TMPRSS2 protein expression levels. #342, #343, and #344 indicate experiments 1, 2, and 3, respectively. N.D., not determined. (C) Left, time courses of SARS-CoV-2 infection and replication in Spheroids^ACE-TMPRSS2S were examined with RT-qPCR targeting the SARS-CoV-2 nucleocapsid. Bars and error bars show geometric mean values ± standard deviations (SD); points represent data from individual microplate wells (n = 6). Right and bottom, representative morphological images from six independently conducted experiments are shown. Right, immunocytostaining to detect SARS-CoV-2 protein with a convalescent IgG fraction that was isolated from serum of a convalescent COVID-19 individual (top) (SARS-CoV-2 antigens, ACE2, and nuclei are indicated in green, red, and blue, respectively) and TEM observation (bottom) (arrows indicate virus particles) were conducted. Bottom, no cytopathic effect on either infected spheroid line was observed with phase-contrast microscopy. (D) Pretreatment and coincubation with anti-human ACE2 MAb with Spheroids^ACE-TMPRSS2S blocked SARS-CoV-2 infection. Bars and error bars show geometric mean values ± SD; points represent data from individual microplate wells (n = 3). Details of the methods are described in the supplemental material.

Here, employing our previously published simplified suspension culture method to obtain large numbers of islet cells/spheroids for transplantation (5), we newly generated two human induced pluripotent stem cell (hiPSC)-derived spheroid lines, termed Spheroids_4M^ACE2-TMPRSS2 and Spheroids_15M63^ACE2-TMPRSS2, both of which express angiotensin-converting enzyme 2 (ACE2) and transmembrane protease serine 2 (TMPRSS2), which are critical for severe acute respiratory syndrome coronavirus 2 (SARS-CoV-2) infection. Both spheroids are highly susceptible to SARS-CoV-2 infection, and two representative anti-SARS-CoV-2 agents, remdesivir and 5h (an inhibitor of SARS-CoV-2's main protease), potently inhibit the infectivity and replicability of SARS-CoV-2 in the spheroids in a dose-dependent manner, suggesting that these induced spheroids should serve as valuable target cells for the evaluation of anti-SARS-CoV-2 activity (6).

In order to induce hiPSC-derived spheroids, we employed the four-step differentiation induction method and the suspension culture method we previously described (5). We successfully induced hiPSCs using the four-step differentiation method, which formed spheroids 10 to 200 μm in diameter by day 58. (Fig. 1A) The relative RNA expression levels of ACE2 and TMPRSS2 were estimated with reverse transcription quantitative PCR (RT-qPCR) based on the expression levels in the normal human lung. The expression level

of ACE2 increased with differentiation induction, while that of TMPRSS2 was found to be virtually constant from differentiation days 37 to 65 (Fig. 1B). Immunocytostaining of ACE2 and TMPRSS2 on day 58 demonstrated their expression and distribution both on the cell membrane and in the cytoplasm (Fig. 1B). Before validating the permissiveness of differentiated cells to SARS-CoV-2, we examined the differentiated cells in more detail. Single-cell RNA sequence analysis revealed that differentiated cells were categorized into 8 and 9 clusters in Spheroids$_{4M}$$^{ACE2-TMPRSS2}$ and Spheroids$_{15M63}$$^{ACE2-TMPRSS2}$, respectively (Fig. S1A in the supplemental material). We also examined the expression profiles of ACE2 and TMPRSS2 with dot plotting, which showed that the two genes encoding the two proteins were expressed in cluster 1 (Fig. S1A). Furin, one of the cellular proteases affecting the viral spike protein, which was important for SARS-CoV-2 infection, was expressed in cluster 1 (Fig. S1A). NKX2.1, a key transcription factor for differentiation of lung cell lineages, was barely expressed in cluster 1 (data not shown). Two intestinal-cell markers, VILLIN1 and CDX2, were considerably expressed in cluster 1, while neuropilin-1 (NRP1), which is reportedly abundant in the respiratory epithelium and potentiates SARS-CoV-2 infectivity through binding to furin-cleaved substrates (7), was slightly expressed in cluster 1 (Fig. S1A). The expression of VILLIN1, CDX2, and FURIN was confirmed by RT-qPCR. The protein expression of VILLIN1 was also corroborated with immunocytochemistry (Fig. S1B). Moreover, we observed the presence of microvilli and junctional complexes on the surface of and between differentiated cells, respectively, as assessed with transmission electron microscopy (TEM) (Fig. S1C). These data, taken together, strongly suggested that the differentiated cells had characteristic features of intestinal cells rather than alveolar type II cells.

Spheroids$_{4M}$$^{ACE2-TMPRSS2}$ and Spheroids$_{15M63}$$^{ACE2-TMPRSS2}$ were then exposed to parental SARS-CoV-2 (strain WK-521, PANGO lineage A; GISAID identifier [ID] EPI_ISL_408667) (SARS-CoV-2$^{WK-521}$) at a multiplicity of infection (MOI) of 0.05 in Eppendorf microtubes and continuously incubated for 3 h. Thereafter, cells were washed with culture medium 3 times and then evenly plated on 96-well microtiter culture plates (Corning 96-well clear, flat bottom, ultra-low-attachment microplates) at a density of ~2 × 10$^5$ cells/well. The supernatants were then collected in the time course of days zero, 1, 2, and 3, and the amounts of the virus in the supernatants were quantified using RT-qPCR. On day 3, spheroids were collected and the localizations of SARS-CoV-2-infected cells and ACE2-positive cells were examined with immunocytostaining (Fig. S2A). The numbers of viral copies in the supernatants, which were quantified using RT-PCR, increased over time with no significant cytopathic effect (CPE) in either spheroid line (Fig. 1 C). In both Spheroids$_{4M}$$^{ACE2-TMPRSS2}$ and Spheroids$_{15M63}$$^{ACE2-TMPRSS2}$, SARS-CoV-2-infected spheroids were found to be diversely distributed, suggesting that their levels of susceptibility to SARS-CoV-2 infection differed substantially (Fig. 1C). Also, it is reported that the expression of ACE2 is downregulated upon SARS-CoV-2 infection (8). Thus, it is likely that the reduced staining with anti-hACE2 antibody observed on the surface of SARS-CoV-2-infected cells was due to the secondary effect ensuing from SARS-CoV-2 infection. The production of viral particles was also confirmed by transmission electron microscopy (TEM) in both spheroid lines (Fig. 1C). Moreover, a blocking test using an ACE2-blocking monoclonal antibody was conducted to confirm that the ACE2 molecules expressed on the surface of the spheroid cells were functionally robust, resulting in SARS-CoV-2 infection. The SARS-CoV-2 infection was inhibited by an anti-human ACE2 (hACE2) monoclonal antibody (AC384) in a dose-dependent manner (Fig. 1D and Fig. S2B), strongly suggesting that the infection of the two spheroids by SARS-CoV-2 was mediated by ACE2 expressed on those cells.

We then asked if two representative anti-SARS-CoV-2 agents, remdesivir (RDV) (9) and 5h (10), blocked the infectivity and replication of SARS-CoV-2 in the spheroids (Fig. 2A). RDV is an inhibitor of RNA-dependent RNA polymerase, while 5h is an inhibitor of the main protease of SARS-CoV-2, each exerting antiviral activity against SARS-CoV-2. First, African green monkey kidney epithelial cell-derived Vero E6 cells were exposed to SARS-CoV-2, seeded, and cultured for 3 days. When the culture supernatants were collected and RT-qPCR was performed as previously described (10), both RDV and 5h

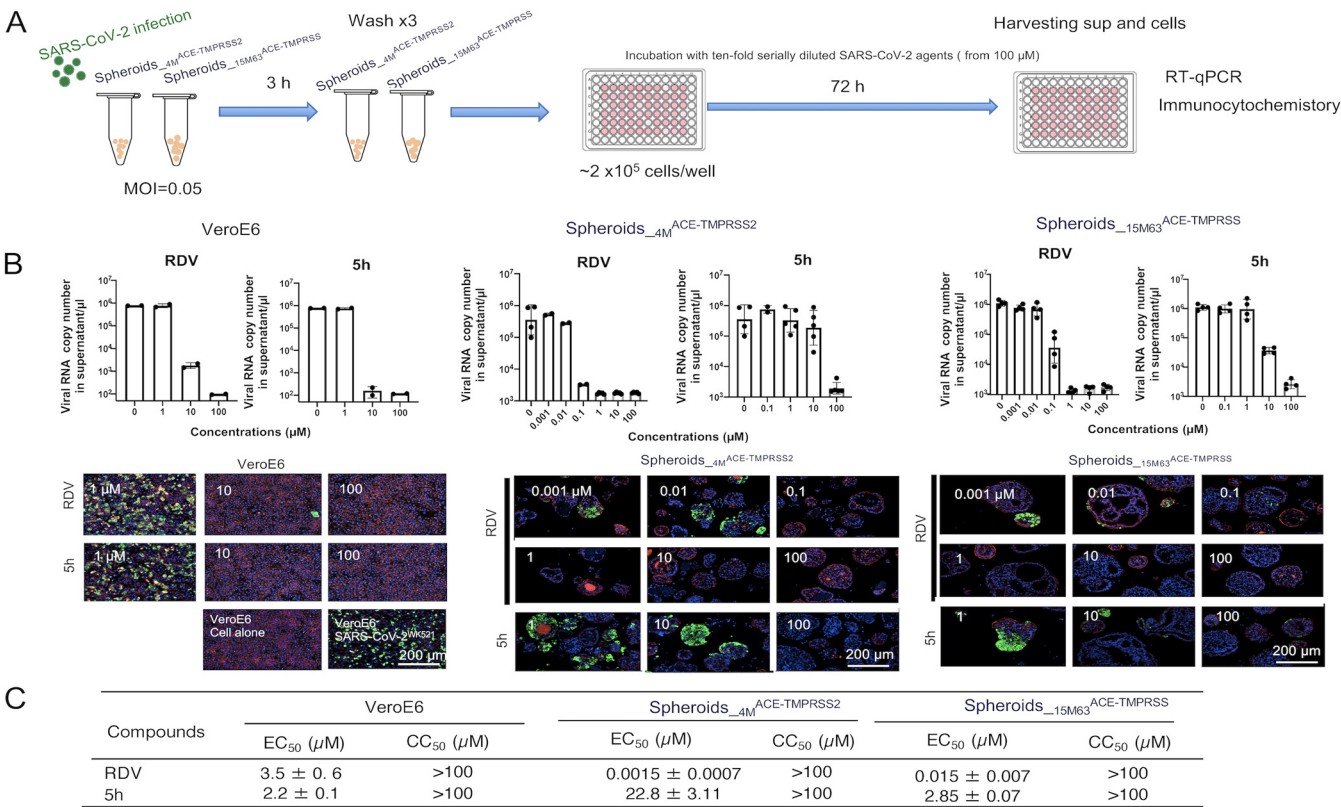

**FIG 2** Evaluation of anti-SARS-CoV-2 agents using Spheroids^ACE-TMPRSS2. The two induced spheroid lines, Spheroids_4M^ACE-TMPRSS2 and Spheroids_15M63^ACE-TMPRSS, were exposed to SARS-CoV-2^WK521 at a multiplicity of infection (MOI) of 0.05. Vero E6 cells were also exposed to SARS-CoV-2^WK-521 (MOI of 0.05) as a control experiment. Three hours postexposure, the virus was washed out three times with culture medium and the cells seeded in 96-well microtiter culture plates at a density of $2 \times 10^6$ cells/well and incubated with or without test compounds for 72 h. On day 3 postexposure, culture supernatants were collected for quantitative SARS-CoV-2 RNA analysis using RT-qPCR, while cells were fixed and subjected to quantitative analysis using immunocytostaining. $EC_{50}$ values were calculated with the obtained data using RT-qPCR. (A) Scheme of the experimental setup. sup, supernatant. (B) Top, inhibition of SARS-CoV-2 infection in Spheroids^ACE-TMPRSS2S and Vero E6 cells was examined with RT-qPCR. Bars and error bars show geometric mean values ± SD; points represent data from individual microplate wells ($n = 4$). Bottom, representative immunocytostaining images from four independently conducted experiments are shown. SARS-CoV-2 antigens, F-actin, and nuclei are indicated in green, red, and blue, respectively. (C) $EC_{50}$ values and $CC_{50}$ values of anti-SARS-CoV-2 agents in Spheroids^ACE-TMPRSS2S were determined using RNA-qPCR and the water-soluble tetrazolium salt (WST-8) based cell viability assay, respectively. Data are mean values ± SD ($n = 4$). Details of the methods are described in the supplemental material.

clearly blocked the infectivity and replication of the virus (Fig. 2B). Comparable to the results in Vero E6 cells, the two agents blocked the infectivity and replication of SARS-CoV-2 in the spheroid lines in a dose-dependent manner. The appearance of SARS-CoV-2-antigen-positive cells was also suppressed, as assessed with immunocytostaining (Fig. 2B). Of note, the antiviral-activity levels of RDV were substantially different between Vero E6 cells and spheroids, with 50% effective concentration ($EC_{50}$) values of 0.0015 $\mu$M in Spheroids_4M^ACE2-TMPRSS2, 0.015 $\mu$M in Spheroids_15M63^ACE2-TMPRSS2, and 3.5 $\mu$M in Vero E6 cells. In the case of 5h, the $EC_{50}$ values in Spheroids_4M^ACE2-TMPRSS2 and Spheroids_15M63^ACE2-TMPRSS2 differed by about 1 log, but the $EC_{50}$ values were comparable in Spheroids_15M63^ACE2-TMPRSS2 and Vero E6 cells (both ~2 $\mu$M). Both antiviral agents had 50% cytotoxic concentration ($CC_{50}$) values of >100 for spheroids, indicating that those agents exerted anti-SARS-CoV-2 activity at concentrations that did not cause cytotoxicity (Fig. 2C). Regarding this, RDV is a prodrug of an antiviral adenosine analogue (GS-5734) that is thought to be intracellularly triphosphorylated to exert its antiviral activity against SARS-CoV-2 (9, 11). Thus, it is plausible that the efficiency of the anabolic phosphorylation of GS-5734 is greater in both Spheroids_4M^ACE2-TMPRSS2 and Spheroids_15M63^ACE2-TMPRSS2 than in Vero E6 cells. On the other hand, the potency of 5h was greater in Vero E6 cells and Spheroids_15M63^ACE2-TMPRSS2 than in Spheroids_15M63^ACE2-TMPRSS2, suggesting that the permeability of 5h is greater, resulting in higher intracellular concentrations of 5h in the former two cell types than in the latter cell type (12).

A caveat in the present study is that the properties of the cells that consist of the two

Spheroids$^{ACE2-TMPRSS2}$ are not exactly those of respiratory cells *per se*, in which SARS-CoV-2 infection reportedly occurs (13). However, in the present work, we definitely established a method for inducing ACE2/TMPRSS2-double-positive hiPSC-derived spheroids that are susceptible to SARS-CoV-2 and apparently suitable for detecting antiviral activity of agents against SARS-CoV-2. The present method should help construct a more suitable and efficient anti-SARS-CoV-2 drug evaluation system.

## SUPPLEMENTAL MATERIAL

Supplemental material is available online only.

**SUPPLEMENTAL FILE 1**, PDF file, 3.4 MB.

## ACKNOWLEDGMENTS

This work was supported in part by the National Center for Global Health and Medicine Research Institute (grant number 20A2014 to H.O. and grant number 20A2001D to H.M.) and the Japan Agency for Medical Research and Development (AMED) (grants number JP20fk0108257 and JP20fk0108510 to H.M.) and in part by the Intramural Research Program of the Center for Cancer Research, National Cancer Institute, National Institutes of Health (H.M.). These funding sources were not involved in study design; collection, analysis, and interpretation of data; the writing of the report; or the decision to submit the paper for publication.

We are grateful to Chinatsu Oyama in the communal laboratory of NCGM Research Institute for her technical support.

We declare that we have no known competing financial interests or personal relationships that could have appeared to influence the work reported in this paper.

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
