## [Reviewer comments · Microbiology Spectrum]

Microbiology Spectrum

Generation of ACE2/TMPRSS2-double-positive-hiPSC-derived Spheroids for Anti-SARS-CoV-2 drug evaluation

Nobuyo Higashi-Kuwata, Shigeru Yabe, Satsuki Fukuda, Junko Nishida, Miwa Tamura-Nakano, Shin-ichiro Hattori, Hitoshi Okochi, and Hiroaki Mitsuya

Corresponding Author(s): Nobuyo Higashi-Kuwata, National Center for Global Health & Medicine Research Institute

Review Timeline:

Submission Date:

October 6, 2022

Accepted:

October 10, 2022

Editor: Takamasa Ueno

Reviewer(s): The reviewers have opted to remain anonymous.

Transaction Report:

DOI: <https://doi.org/10.1128/spectrum.03490-22>

October 10, 2022

Dr. Nobuyo Higashi-Kuwata
National Center for Global Health & Medicine Research Institute
Tokyo
Japan

Re: Spectrum03490-22 (Generation of ACE2/TMPRSS2-double-positive-hiPSC-derived Spheroids for Anti-SARS-CoV-2 drug evaluation)

Dear Dr. Nobuyo Higashi-Kuwata:

Your manuscript has been accepted, and I am forwarding it to the ASM Journals Department for publication. You will be notified when your proofs are ready to be viewed.

Sincerely,

Takamasa Ueno
Editor, Microbiology Spectrum

Journals Department
Supplemental Material FOR Publication: Accept